# A *Medicago truncatula* Autoregulation of Nodulation Mutant Transcriptome Analysis Reveals Disruption of the SUNN Pathway Causes Constitutive Expression Changes in Some Genes, but Overall Response to Rhizobia Resembles Wild-Type, Including Induction of *TML1* and *TML2*

Elise L. Schnabel [1] , Suchitra A. Chavan [2], Yueyao Gao [1] , William L. Poehlman [3], Frank Alex Feltus [1,4,5] and Julia A. Frugoli [1,*]

1    Department of Genetics and Biochemistry, Clemson University, Clemson, SC 29634, USA
2    Leidos, Inc., Atlanta, GA 30345, USA; schavanclemson@gmail.com
3    Sage Bionetworks, Seattle, WA 98121, USA
4    Biomedical Data Science and Informatics Program, Clemson University, Clemson, SC 29634, USA
5    Clemson Center for Human Genetics, Clemson University, Greenwood, SC 29636, USA
*    Correspondence: jfrugol@clemson.edu

**Abstract:** Nodule number regulation in legumes is controlled by a feedback loop that integrates nutrient and rhizobia symbiont status signals to regulate nodule development. Signals from the roots are perceived by shoot receptors, including a CLV1-like receptor-like kinase known as SUNN in *Medicago truncatula*. In the absence of functional SUNN, the autoregulation feedback loop is disrupted, resulting in hypernodulation. To elucidate early autoregulation mechanisms disrupted in SUNN mutants, we searched for genes with altered expression in the loss-of-function *sunn-4* mutant and included the *rdn1-2* autoregulation mutant for comparison. We identified constitutively altered expression of small groups of genes in *sunn-4* roots and in *sunn-4* shoots. All genes with verified roles in nodulation that were induced in wild-type roots during the establishment of nodules were also induced in *sunn-4*, including autoregulation genes *TML2* and *TML1*. Only an isoflavone-7-O-methyltransferase gene was induced in response to rhizobia in wild-type roots but not induced in *sunn-4*. In shoot tissues of wild-type, eight rhizobia-responsive genes were identified, including a MYB family transcription factor gene that remained at a baseline level in *sunn-4*; three genes were induced by rhizobia in shoots of *sunn-4* but not wild-type. We cataloged the temporal induction profiles of many small secreted peptide (MtSSP) genes in nodulating root tissues, encompassing members of twenty-four peptide families, including the CLE and IRON MAN families. The discovery that expression of *TML2* in roots, a key factor in inhibiting nodulation in response to autoregulation signals, is also triggered in *sunn-4* in the section of roots analyzed, suggests that the mechanism of TML regulation of nodulation in *M. truncatula* may be more complex than published models.

**Keywords:** autoregulation of nodulation; *Medicago truncatula*; RDN1; small secreted peptides; SUNN; TML

## 1. Introduction

Legumes can grow on nitrogen-poor soil because of the ability to establish a symbiosis with nitrogen-fixing soil bacteria. The legume–rhizobia symbiosis, in which rhizobia inhabit the roots of legume plants and fix nitrogen from the atmosphere in exchange for carbon from photosynthesis, is an example of complex signaling between two very different species over both space (soil, root, shoot) and time. The time from rhizobial encounter in the soil to established nitrogen fixing nodules ranges from 8 to 20 days, depending on species and conditions. Rhizobia secrete Nod factors in response to flavonoids exuded from legume

roots; then, in the distal root area of emerged root hairs known as the maturation zone, the root hairs curl and entrap the bacteria, and calcium pulses triggered by the interaction rapidly begin altering plant gene expression. An infection thread develops around the dividing rhizobia and passes through the outer cortex to already dividing cells in the inner cortex, where the bacteria are released from infection threads into the developing nodule [1–3]. At each step in establishing symbiosis, there are checkpoints that must be cleared, as evidenced by the number of plant mutants in nodulation arrested at different developmental stages, such as infection thread curling, penetration of the thread through cell division of the rhizobia, rearrangement of the cellular components to allow the thread to pass through the cortex, establishment of the nodule meristems, colonization of the rhizobia, and establishment of nitrogen fixation [3]. A large percentage of even compatible interactions is arrested in the outer cortex for reasons not yet known, and this occurs in the first 24–48 h after infection [4].

Once the plant has committed to the nodulation process, it controls the number of nodules that form and monitors the nitrogen output, as there is an energy cost to the process of 12 to 16 g of carbon per gram of nitrogen fixed [5], and intermittent or excess nitrogen offers little advantage to the plant. Multiple genes have been shown to control nodule number, including genes in the autoregulation of nodulation (AON) pathway. This systemic pathway is initiated by the interaction of roots with rhizobia followed by transport of newly synthesized mobile peptide signals (CLEs) to a receptor complex in the shoot. In coordination with other pathways monitoring nutrient status, perception of the signal in the shoots causes changes in cytokinin and auxin flux and reduced transport of a mobile miRNA (*miR2111*) to roots, which is proposed to allow accumulation of the nodulation, inhibiting proteins that are the targets of the miRNA [3,6]. The effects of AON can be detected in *M. truncatula* before nodules have fully developed, within 48 h of inoculation [7], suggesting that AON signaling is happening simultaneously with nodule development signaling.

Mutations in two AON components give similar phenotypes. The *SUPERNUMERARY NODULES* (*SUNN*) gene encodes part of the receptor complex, a CLAVATA1-like leucine-rich repeat receptor-like kinase, a key regulator of nodule number acting as a shoot receptor for the mobile signaling CLE peptides induced in roots by rhizobia as well as mycorrhizae and nitrogen [8–10]. While all *sunn* alleles have short roots and auxin transport defects, the *sunn-4* allele has a stop codon very early in the coding sequence, and this null allele has a stronger hypernodulation phenotype than the original *sunn-1* allele, which harbors a kinase domain missense mutation [10,11]. The *ROOT DETERMINED NODULATION1* (*RDN1*) gene encodes a hydroxyproline O-arabinosyltransferase that modifies one of the root-to-shoot signaling peptides, MtCLE12, enhancing transport and/or reception by the SUNN kinase [12]. The *rdn1-2* allele has an insertion within the gene that greatly reduces the level of mature *RDN1* mRNA and affects the AON pathway upstream of SUNN [12,13].

To expand discovery of genes contributing to the developmental and nodulation phenotypes of AON mutants, we focused on early time points where AON and nodule initiation are happening simultaneously. We explored temporal gene expression differences between the sequenced wild-type (A17) of the model indeterminate nodulating legume *Medicago truncatula* and mutants with lesions in *SUNN* (*sunn-4*) and the *RDN1* (*rdn1-2*). We combined a harvesting procedure [14] incorporating simultaneous inoculation of all plants in an aeroponic system [15] with harvest of the zone of developing nodules tracked via root growth. In our laboratory experience, nodulation progresses more uniformly and more rapidly in this system than on plates or in pots, and we presume this is due to the apparatus simultaneously spraying the aerosolized solution of rhizobia on all plants. The transcriptome data generated from the specific area of the root responding to our inoculation from these three genotypes identified a small collection of genes from those differentially expressed in both wild-type and AON mutants responding to rhizobia, which showed constitutively altered expression in root and shoot segments of the AON mutants. Examining the gene expression in nodulating root tissues during a time course covering the

first 72 h of interaction with rhizobia, as well as the shoots of these plants, we found that the AON mutants have a transcriptional response to rhizobia similar to wild-type, including upregulation of many known nodulation pathway genes in roots and a small number of genes in the shoots. We found only two genes identified as responding to rhizobia in wild-type that failed to also respond in *sunn-4*, one in roots and one in shoots, and these two genes displayed a moderated response in *rdn1-2*.

Unexpectedly, genes upregulated in both wild-type and AON mutant roots included *TML1* and *TML2*, which encode nodulation-inhibiting proteins in the AON pathway whose expression has been proposed to be increased by a SUNN-dependent decrease in *miR2111* levels in roots following nodule initiation using a split root system [16]. Our observed induction of *TML1* and *TML2* in *sunn-4* root segments is not in agreement with a model in which increased *TML* levels in this area of the root at 48 h post-inoculation is sufficient to control nodule number, suggesting that additional factors dependent on SUNN function may be required for the later steps of AON or in other parts of the root.

## 2. Materials and Methods

### 2.1. Plant Growth and Tissue Sampling

Seeds of *Medicago truncatula* lines A17, *sunn-4*, and *rdn1-2* harvested from plants grown in our greenhouse, were scarified in concentrated sulfuric acid (93–98%) by vortexing for 8 min in 50 mL sterile plastic conical tubes. The seeds were then rinsed in distilled water five times and imbibed for 3 to 4 h in water with gentle shaking at room temperature. Following imbibition, the seeds were plated by suspending over water on the lids of petri dishes and vernalized for 2 d in the dark at 4 °C before being germinated overnight in darkness at 22 °C. Seedlings were loaded onto an aeroponic growth apparatus and grown under a 14 h light/10 h dark cycle at 22 °C in nodulation medium with no supplemental nitrogen [15]. After 3 d of growth, samples of 20 plants per genotype (t = 0 h) were collected (3.5 h after initiation of daily light) and processed as described below and in Figure 1A. For inoculations, 150 $OD_{600}$ Units ($12 \times 10^{10}$ CFUs) of *Sinorhizobium/Ensifer medicae* ABS7 resuspended in 40 to 100 mL of nodulation medium were added to the growth apparatus immediately after collection of the 0 h samples. Additional samples of 20 plants per genotype were collected 12, 24, 48, and 72 h later. Nodule primordia began the first cell divisions by 24 h post-inoculation (hpi) (Figure 1B) and emerged nodules were visible on the root by 72 hpi.

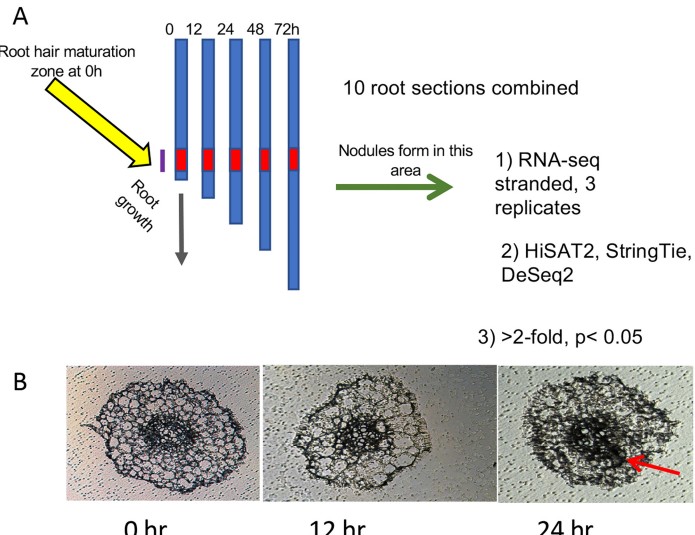

**Figure 1.** Diagram of experimental procedure designed to increase signal to noise ratio: (**A**) Twenty plants per genotype (t = 0 h) were collected and rhizobia added to the growth apparatus immediately

after collection of the 0 h samples. Additional samples of 20 plants per genotype were collected at 12, 24, 48, and 72 h after the 0 h samples. Ten plants from each collection were used to determine average root length and 2 cm segments representing the zone of development of the first nodules (red zone) were collected from the remaining 10 plants. At 0 h, this region started 1 cm from the root tip, where the first full-length root hairs were present. At later time points, this region was determined by calculating the average root growth since $t = 0$ and adding this distance to 1 cm. (**B**) Cross-sections of roots harvested during the first three time points. Cell division for nodule formation was occasionally observed at 24 h (red arrow). By 48 h, nodule cell division was observed in all plants.

Roots of ten plants per sample were measured to determine average root length for the following procedure. From the remaining ten plants, 2 cm segments representing the zone of development of the first nodules were collected. For 0 hpi samples, this region started 1 cm from the root tip, where the first full-length root hairs were present. For later time points, the region of the first developing nodules was tracked by monitoring the progression of root growth away from the first interacting cells. The region was determined by calculating the average root growth since t = 0 (average root length at t minus average root length at 0 h) and adding this distance to 1 cm, and 2 cm sections were collected upward from that position. This area is referred to as "root segments" throughout the text. Shoots, mid-hypocotyl and up, were collected from plants at the same time points. Collected tissue samples were placed into 1.5 mL tubes and stored at −80 °C.

### 2.2. RNA Preparation, Libraries, and Sequencing

RNA was purified from frozen samples by grinding in liquid nitrogen and using the RNAqueous Total RNA Isolation Kit (Invitrogen). Aliquots of RNA were analyzed for quality and concentration on an Agilent 2100 Bioanalyzer. RNA samples had RIN values between 8.3 and 9.9 for roots and 6.3 and 8.5 for shoots. Libraries for RNAseq were prepared and sequenced by Novogene Co., Ltd. (Beijing, China) from 100 to 1000 ng of total RNA using a stranded kit (Illumina TruSeq Stranded Total RNA Kit or NEB Next Ultra™ II Directional RNA Library Prep Kit for Illumina). The resulting data files contained paired-end sequences (150 bp) that ranged from 18,674,569 to 64,491,795 fragments.

### 2.3. Analysis of Gene Expression

The root dataset consists of 75 libraries, including 60 libraries from this work (three replicates of five time points each for inoculated wild-type A17, *sunn-4* and *rdn1-2* root segments, and uninoculated *sunn-4* root segments) and 15 libraries from uninoculated wild-type A17 root segments generated in the same way and used in two manuscripts on new analysis algorithms [14,17]. The shoot dataset has 60 libraries (three replicates of five time points each for inoculated wild-type A17, *sunn-4*, and *rdn1-2* and uninoculated A17), including 30 from this work for the two mutants and 30 from wild-type previously reported [17]. Read mapping and alignment data for each library is included in Supplemental Dataset 1. Among the genome v4.1 transcripts identified in our analyses were six that have been updated in v5, merging three pairs of transcripts (Medtr7g028557 and Medtr7g028553, Medtr3g065345 and Medtr3g065350, and Medtr0027s0200 and Medtr0027s0180) into three genes (MtrunA17Chr7g0224711, MtrunA17Chr3g0110301, and MtrunA17Chr7g0240781, respectively). All the v4.1 transcripts are listed in the figures, but the v5 annotations were used for determining gene totals.

The datasets were processed with a DESeq2 pipeline using *Medicago truncatula* genome v4.1 as described in [14]. We tested for differential expression at each time point using a cut-off of adjusted *p*-value < 0.05 and minimum fold change of 2. Three pairwise comparisons of gene expression levels were performed at each timepoint (0, 12, 24, 48, and 72 hpi) both for root segments and for shoots to create gene lists for further screening: (1) *sunn-4* (inoculated) versus A17 (inoculated), (2) *rdn1-2* (inoculated) versus A17 (inoculated), and (3) A17 (inoculated) versus A17 (uninoculated). Genes that were flagged as differential between the two 0 h datasets of A17 root segments, between the two 0 h datasets of A17 shoots, and

between two 0 h datasets of *sunn-4* root segments were excluded from further analysis in those tissues to eliminate noise.

To identify genes with constitutively higher or lower expression in the AON mutants, genes from all three comparisons were further screened by assessing expression across all time points with heat map filtering and visual analysis of time course graphs. From 10,299 candidate genes in root segments and 749 in shoots, 32 and 49 genes were identified with consistently higher or lower expression in AON mutant roots and shoots, respectively.

Genes identified in root segments of A17 inoculated versus uninoculated (12, 24, 48, or 72 hpi) and identified as expressed at least two-fold lower in *sunn-4* inoculated versus A17 inoculated (12, 24, 48, or 72 hpi) were also further assessed with heat map filtering and visual analysis of time course graphs. Of 2155 candidate rhizobial response genes in A17 root segments, 477 were flagged by DeSeq2 as lower in expression in inoculated *sunn-4* root segments compared to A17; heat map analysis and visual analysis of time course graphs identified three of these genes with clearly reduced rhizobial response in *sunn-4* roots segments.

To identify rhizobial response genes in shoots, differentially expressed genes from all shoot comparisons were further screened with heat map filtering and visual analysis of time course graphs. From 749 candidate genes, 11 rhizobial response genes of shoots were identified.

Expression changes in selected genes were assayed by quantitative PCR using biological replicates independent of those used for RNAseq analysis. RNA was purified using the E.Z.N.A. Plant RNA Kit (Omega Bio-Tek) from sections of five roots. The iScript cDNA Synthesis Kit (Bio-Rad) was used to synthesize cDNA from 350 ng of RNA, using gene-specific primers from Supplemental Table S1. Relative gene expression was assayed on the iQ5 system (Bio-Rad) using iTaq Universal SYBR Green Supermix (Bio-Rad). Expression levels (fold change) were determined by comparison to the expression of control gene PI4K (Medtr3g091400).

- Functional enrichment analysis was performed with the Medicago Classification Superviewer (http://bar.utoronto.ca/ntools/cgi-bin/ntools_classification_superviewer_medicago.cgi accessed multiple times in July 2020) using the default settings and a significance threshold of $p < 0.05$ [18].

## 3. Results

### 3.1. Constitutively Altered Gene Expression in sunn-4 Roots and Shoots

We compared gene expression in *sunn-4* and *rdn1-2* to wild-type (A17) to identify genes that were always different regardless of time or treatment in the AON mutants. Twenty-seven genes were found to have consistently higher ($n = 15$) or lower expression ($n = 12$) in *sunn-4* root segments compared to wild-type (Figure 2). Nine of these genes were similarly altered in *rdn1-2*, and an additional three were altered in *rdn1-2* only (including *rdn1* itself). Among the eight genes more highly expressed in both AON mutants compared to wild-type root segments were *NF-YA2* (Medtr7g106450), a CAAT-binding transcription factor known to influence nodulation [19,20], and *MtSPL4* (Medtr2g014200), a SQUAMOSA promoter binding protein-like transcription factor. SPLs bind to SBP domain binding sites in promoters [21].

Reduced expression of a subset of *SPLs* in *Lotus japonicus* was observed when *miR156* was overexpressed in roots, and the authors hypothesize that *miR156* directly or indirectly targets *ENOD40*, a gene important to nodule biogenesis [22]. The expression of a gene predicted to encode a 55 amino acid type II membrane protein of unknown function (Medtr2g090685) was lower in both AON mutants. The group of genes was determined to be enhanced for transcription factor ($p = 0.012$, Medicago Classification Superviewer) and transporter activities ($p = 0.007$).

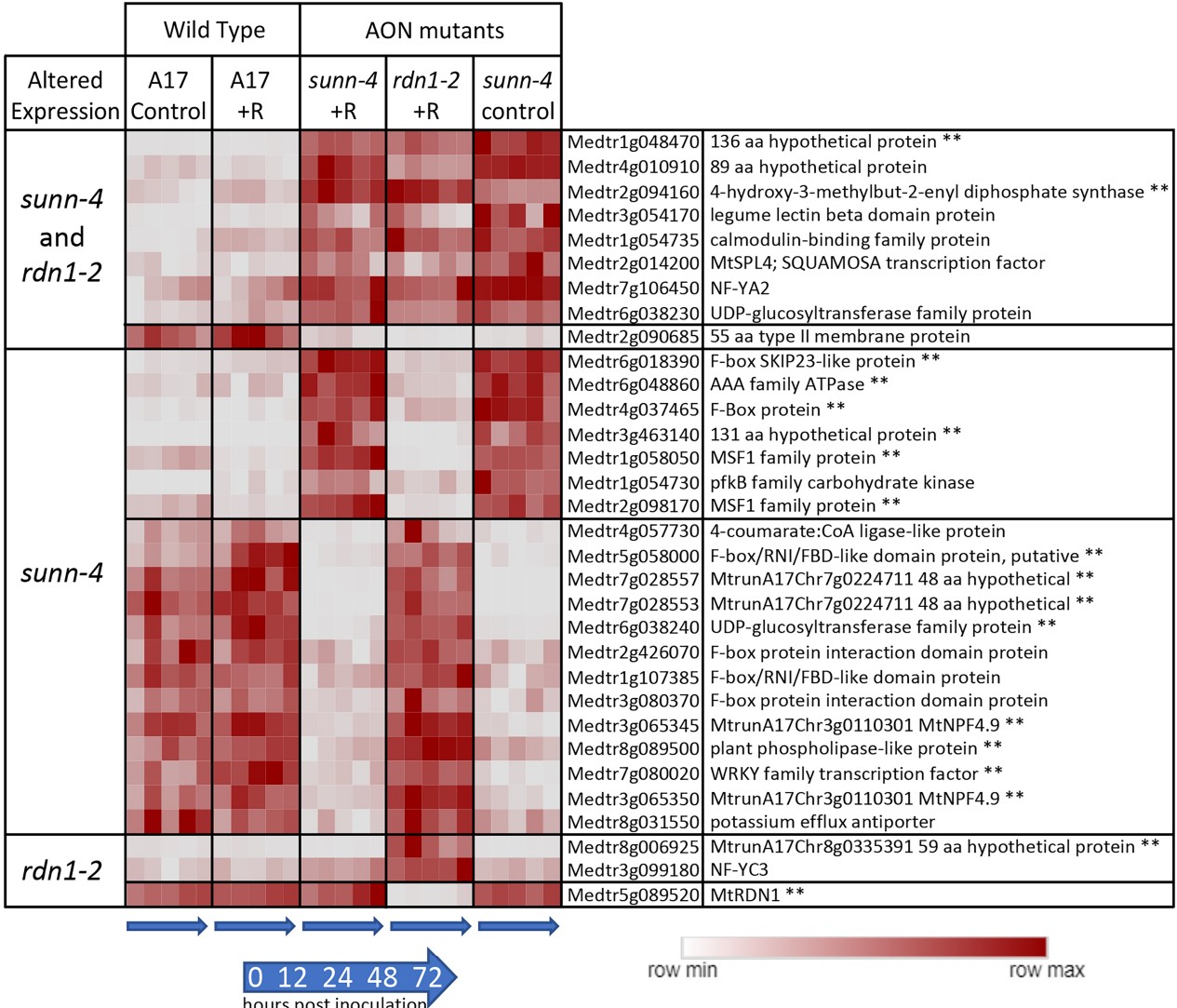

**Figure 2.** Genes with constitutively altered expression in roots of AON mutants *sunn-4* and *rdn1-2*. Heat map of average fragments per kilobase of transcript per million mapped reads (FPKMs) of genes identified by DeSeq2 with altered expression levels in AON mutants compared to wild-type (A17) that were consistent across all times and conditions (control = no rhizobia; +R = with rhizobia). Each row is independently scaled from minimum to maximum values; underlying data are in Supplemental Dataset 2, sheet F1. Expression of some genes was altered in both mutants, while for others the difference was only found in one mutant. Some genes had higher expression in the mutants and some had lower. The geneID (v4) and annotation are given. For five geneIDs, the annotation in v5 better matched the transcript structure; the v5 geneID is also given for these, with two pairs of geneIDs merged into two larger genes in v5. ** Also found to be similarly different in shoots of AON mutants (see Figure 2).

Forty-one genes were found to have consistently higher ($n = 18$) or lower expression ($n = 23$) in *sunn-4* shoots compared to wild-type (Figure 3). Seventeen of these genes were similarly altered in *rdn1-2*, and an additional six were altered in *rdn1-2* only (including *rdn1* itself). Fourteen of the genes were also found among those higher ($n = 8$) and lower ($n = 6$) in roots of *sunn-4*.

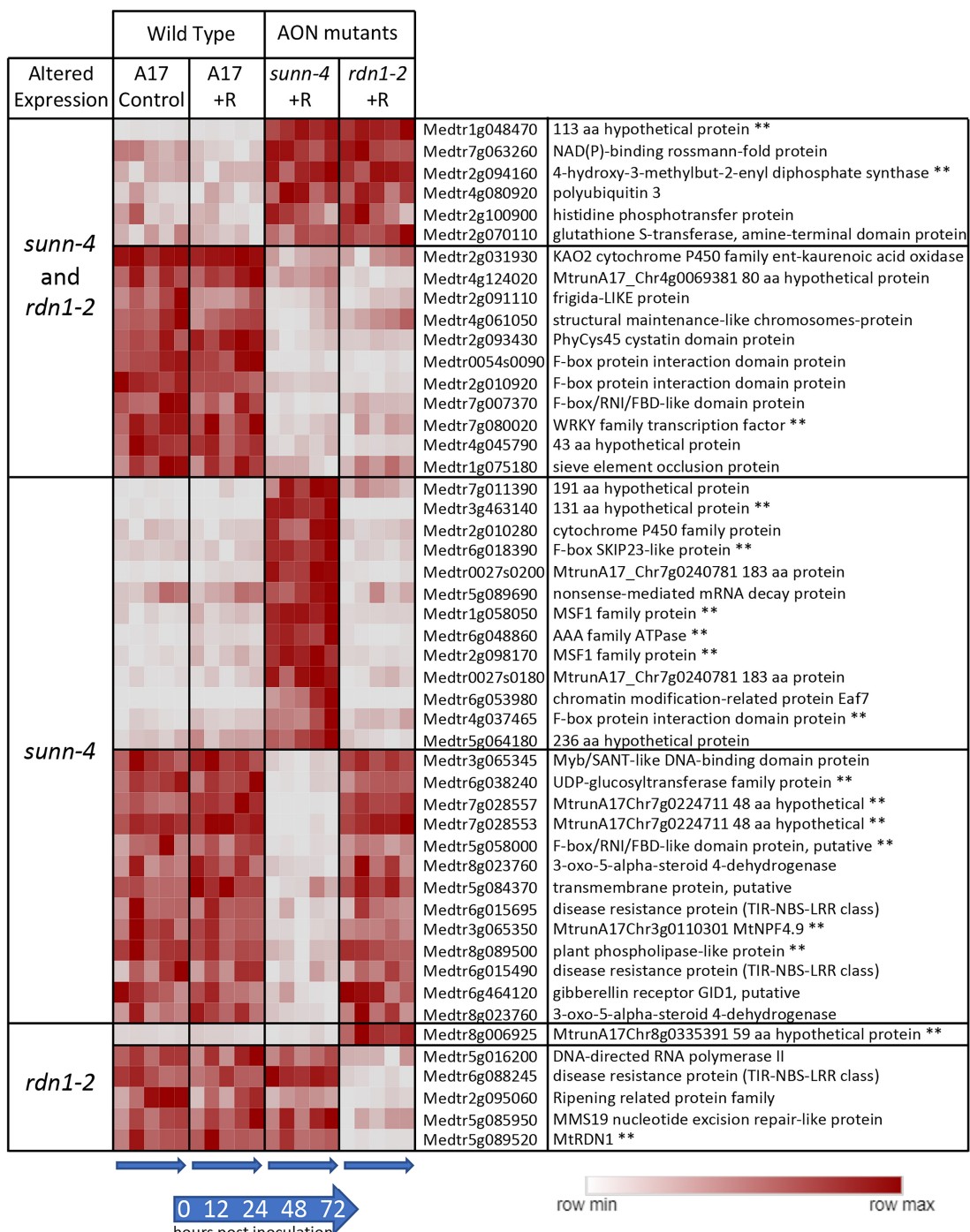

**Figure 3.** Genes with constitutively altered expression in shoots of *sunn-4* and *rdn1-2*. Heat map of average fragments per kilobase of transcript per million mapped reads (FPKMs) of genes identified by DESeq2 with altered expression levels in AON mutants compared to wild-type (A17) that were consistent across all times and conditions (control = no rhizobia; +R = with rhizobia). Each row is independently scaled from minimum to maximum values; underlying data are in in Supplemental Dataset 2, sheet F2. Expression of some genes was altered in both mutants, while for others the difference was only found in one mutant. Some genes had higher or lower expression in the mutants. The geneID (v4) and annotation are given. For seven geneIDs, the annotation in v5 better matched the transcript structure; the v5 geneID is also given for these, with two pairs of geneIDs merged into two larger genes in v5. ** Also found to be similarly different in roots of AON mutants (see Figure 2).

*3.2. Response of Genes to Rhizobia in Roots in Wild-Type and sunn-4*

3.2.1. Nodulation Pathway Genes

We assessed our dataset for the behavior of 207 functionally validated symbiotic nitrogen fixation genes ranging in roles from early nodulation signaling to nitrogen fixation [3]. While not all these genes would be expected to respond to rhizobia, 56 of the 207 genes (27%) had increased expression after inoculation, while two showed lower expression in AON mutants (Figure 4). *RDN1* showed consistently lower expression in *rdn1-2*, as had previously been shown for this mutant [13], and the synaptotagmin gene *MtSYT2* had decreased expression in AON mutants at the 72 hpi time point. *MtSYT2* encodes a synaptotagmin from a family of three in *M. truncatula* shown by localization and RNAi to be involved in formation of the symbiotic interface [23].

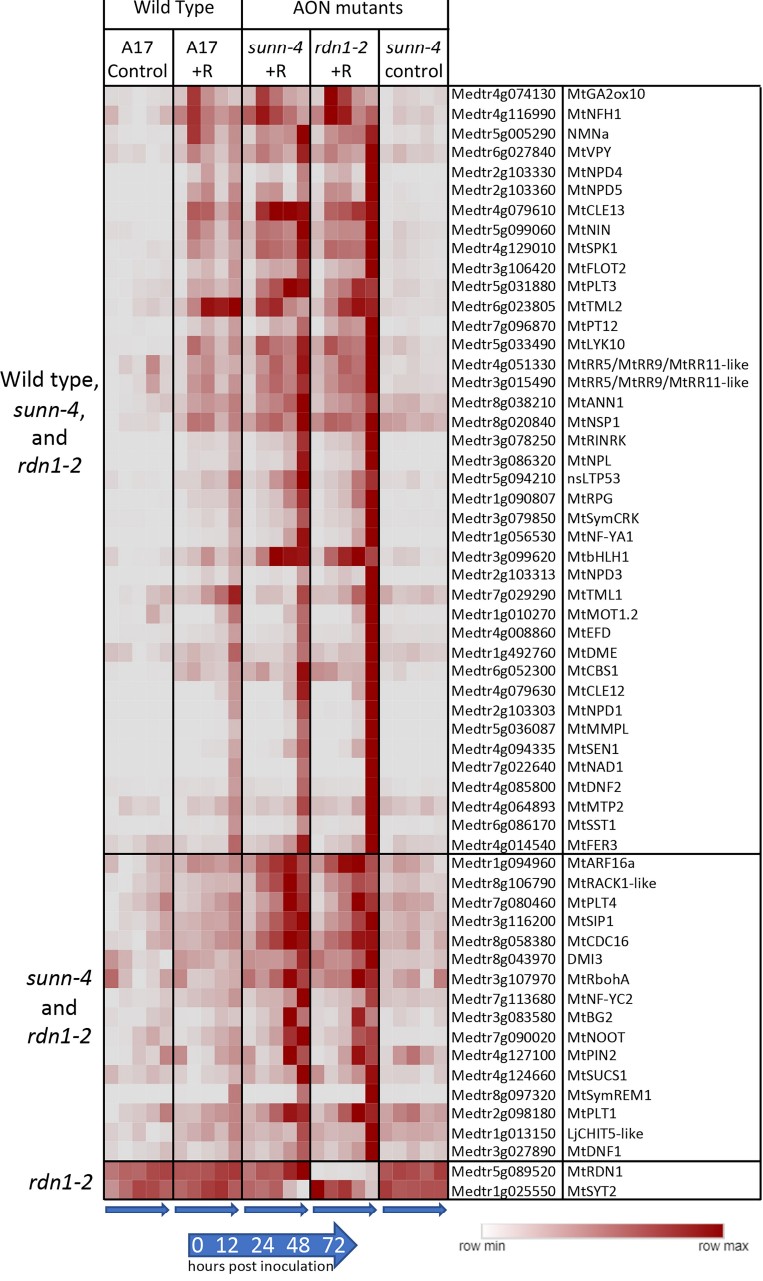

**Figure 4.** Rhizobia-induced expression of nodulation pathway genes in roots of wild-type (A17) and/or AON mutants *sunn-4* and *rdn1-2*. Heat map of average fragments per kilobase of transcript

per million mapped reads (FPKMs) of 58 known nodulation genes with patterns of expression that changed with rhizobial inoculation (+R) or with genotype. Each row is independently scaled from minimum to maximum values; underlying data are in Supplemental Dataset 2, Sheet F3. Induction was detected in all three lines for some genes (n = 40) and for only the AON mutants for others (n = 16). Two genes were altered in *rdn1-2* only.

Forty of the differentially expressed genes in Figure 4 were induced by rhizobia in both wild-type and AON mutants with most genes more highly induced in the mutants. Included among these genes induced in all genotypes are four nodule PLAT domain proteins, which are known to be expressed in nodules [24], but interestingly, unlike *MtNPD1*, which showed increased expression only at 48 and 72 hpi, *MtNPD4* and *MtNPD5* had increased expression by 12 hpi and *MtNPD3* by 24 hpi. Induction of sixteen genes was only seen in the AON mutants and included five genes increasing by 12 hpi and eleven genes by 48 or 72 hpi. Included among genes induced early in AON mutants is the PLETHORA gene *Mt-PLT4*, which has been shown to be expressed in the central areas of nodule meristems [25]. *MtPLT1*, known to be expressed in peripheral areas of nodule meristems, was increased at 48 or 72 hpi timepoints.

### 3.2.2. TMLs

Two genes, *TML1* and *TML2*, whose upregulation in roots is proposed to be a key part of the AON pathway downstream of SUNN, showed an unexpected response to rhizobia in AON mutants (Figure 5). In wild-type and in AON mutants, *TML1* and *TML2* RNA levels increased in response to rhizobia. *TML2* expression was induced by 12 hpi and peaked by 24 hpi in all three genotypes (Figure 5A). For *TML1*, RNA levels started increasing around 24 hpi and continued to rise in wild-type and *rdn1-2*; in *sunn-4*, *TML1* expression began to increase later (Figure 5B). Given the rise in transcript abundance for both genes in *sunn-4*, we tailored the qPCR confirmation of the result to the times of induction. For *TML2* we chose to divide the interval before the increase into smaller fractions to verify our unexpected finding of increased RNA expression by qPCR, rather than repeat the entire time course. Since *TML1* expression rose later in the time course, we repeated the entire time course for *TML1*. The overall patterns of expression for *TML2* (Figure 5C) and *TML1* (Figure 5D) were similar to wild-type in independent samples assayed, with wild-type and *sunn-4* showing increased *TML2* by 8 hpi and *TML1* by 48 hpi; however, the qPCR showed a smaller rise for wild-type for *TML2* and no decrease in *sunn-4* for *TML1*. Taken together, the transcriptome and qPCR confirm small rises in *sunn-4* and wild-type for *TML2* between 0 and 16 h, with a small decrease at 24 h that continues in *sunn-4* but not A17 based on the transcriptome at 72 hpi. *TML1* expression increases over the time course to a similar level in both genotypes.

### 3.2.3. Genes Unresponsive to Rhizobia in *sunn-4* Mutants

A screen for genes induced by rhizobia in wild-type but not in *sunn-4* yielded three genes, one with increased expression over 72 hpi (Figure 6) and two with increased expression by 12 hpi that was then reduced (Figure S1). Following tests of independent samples by qPCR, it was found that an isoflavone 7-O-methyltransferase gene (Medtr7g014510) showed consistently increasing expression in wild-type over the 72 h following inoculation but showed a much lower increase in *sunn-4*. For the other two genes (Medtr2g086390, a b-ZIP transcription factor, and Medtr1g109600, a putative small signaling peptide), the induction that was observed in the RNAseq data was not found in the qPCR data (Figure S1).

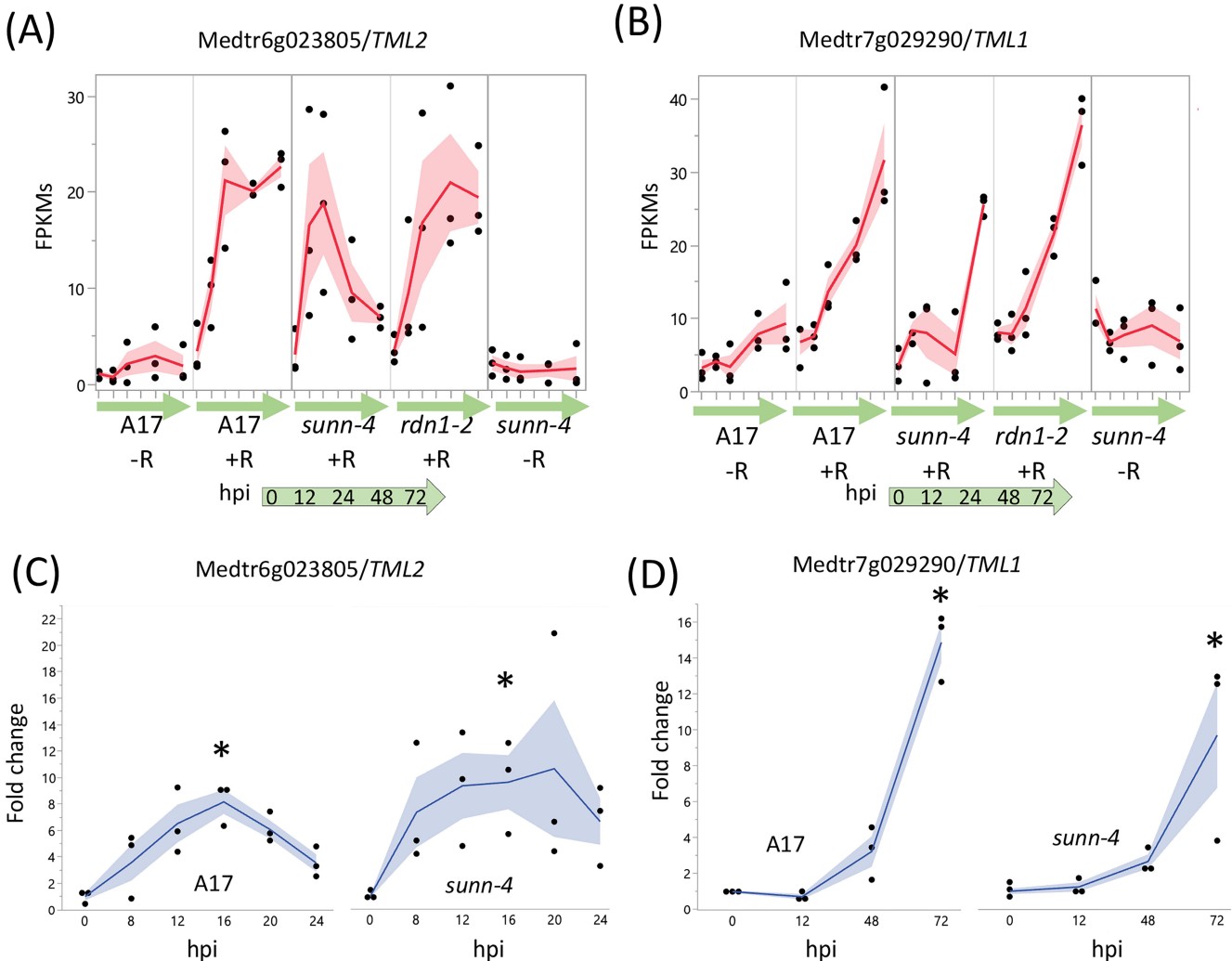

**Figure 5.** AON genes *TML2* and *TML1* are induced by rhizobia in both wild type and AON mutant root segments. RNA-seq data shows early induction in response to rhizobia for *TML2* (**A**) and later induction for *TML1* (**B**). The fragments per kilobase of transcript per million mapped reads (FPKMs) (black dots) and means (red lines) of three biological replicates are shown for time points 0 through 72 h post-inoculation (hpi) for uninoculated (−R; wild-type A17 and *sunn-4*) and inoculated (+R; wild-type, *sunn-4*, and *rdn1-2*) root segments. qPCR verified the induction of *TML2* (**C**) and *TML1* (**D**) in both A17 and *sunn-4*. Expression levels were significantly higher in both wild-type and *sunn-4* at 16 hpi for *TML2* and at 72 hpi for *TML1* (*, $p < 0.05$; Kruskal–Wallis test with Bonferroni correction). The relative expression of three biological replicates (black dots = data points; blue line = means) of these genes is shown. Shading shows the standard error of the mean.

### 3.2.4. Induction of Small Signaling Peptide Genes in Roots

Members of the CLE peptide gene family were previously identified as playing an important role in nodulation regulation [26,27]. We found 170 peptide-encoding genes from the *Medicago truncatula* Small Secreted Peptide Database [28] that showed a rhizobia-induced increase of expression in wild-type and/or AON mutants (Table 1). Twenty-four peptide families are represented by these genes, including NCR peptides (n = 46) and CLE peptides (n = 10). Expression of some peptide genes began increasing by 12 hpi (n = 23), such as seven plant defensins, while others were induced at later times (by 24 hpi, n = 9; by 48 or 72 hpi, n = 138), including all fourteen members of the IRON MAN peptide family [29]. Interestingly, although NCR peptides are known to accumulate in nodules to aid bacterial differentiation [30], NCR*150* (Medtr6g466410) showed a transient increase in expression

at 12 and 24 hpi (Figure S2), suggesting a role for NCR150 in a non-nodule tissue. High levels of *CLE12* and *CLE13* have been documented at 3 to 15 days post-inoculation, with *CLE13* levels increasing earlier than CLE12 [27]; we found that *CLE13* is induced by 12 hpi, while *CLE12* levels do not begin to rise significantly until 48 to 72 hpi (Figure S4). Early expression of *CLE13* in AON mutants was as in wild-type, although by 72 hpi levels in mutants were two-fold higher, while induction of *CLE12* in AON mutants was stronger, consistent with the data in [12].

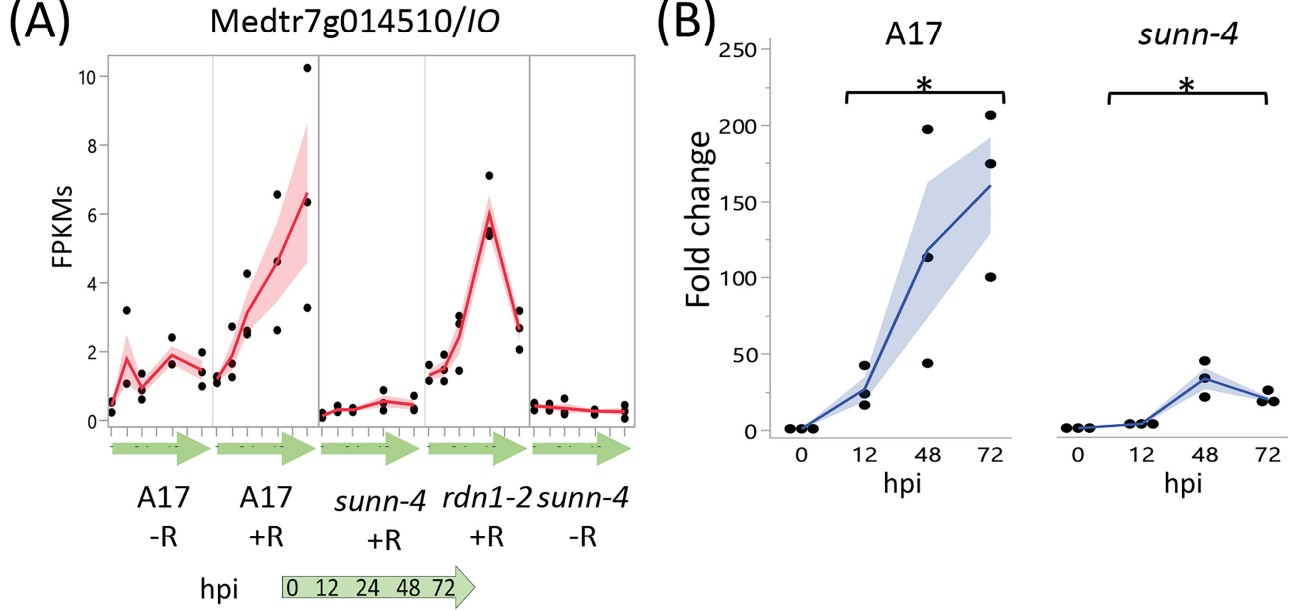

**Figure 6.** Strong induction of an Isoflavone 7-O-Methyltransferase (Medtr7g014510) in nodulating root segments of wild type plants is not seen in *sunn-4plants*. (**A**) fragments per kilobase of transcript per million mapped reads (FPKMs) (black dots) and means (red lines) from three biological replicates for Medtr7g014510 from RNA-seq of wild-type and the AON mutants *sunn-4* and *rdn1-2* over the first 72 h post-inoculation (hpi) with rhizobia (+R) compared to uninoculated controls (−R). (**B**) qPCR analysis of Medtr7g014510 in wild-type and *sunn-4* showing expression levels relative to 0 h. Blue line is mean. Post-inoculation times points were significantly higher than the 0 h samples, although the extent of induction was 3- to 9-fold less in *sunn-4* (*, $p < 0.05$; Kruskal–Wallis test). Shading is the standard error of the mean.

**Table 1.** Small secreted peptide (MtSSP) genes induced by rhizobia in the maturation zone of wild-type or autoregulation mutants *sunn-4* and *rdn1-2* during nodule development. **Bold** = induction only as DEGs in AON mutants.

| Peptide Family | Induced Genes /Total Genes (v4 Genome) | Increase 12 hpi | Increase 24 hpi | Increase 48–72 hpi | Peptide Family Description |
|---|---|---|---|---|---|
| BBPI | 1/16 | | | **BBPI16** | Bowman–Birk Peptidase Inhibitor |
| CAPE | 4/21 | | *CAPE1* | *CAPE2, CAPE16, CAPE18* | CAP-derived Peptide |
| CEP | 1/10 | | | *CEP14* | C-terminally Encoded Peptide |
| CLE | 10/46 | *CLE13, CLE53* | **CLE29**, *CLE35* | *CLE12,* **CLE34,** **CLE37, CLE41,** *CLE44, CLE45* | Clavata/Embryo Surrounding Region |

**Table 1.** *Cont.*

| Peptide Family | Induced Genes /Total Genes (v4 Genome) | Increase 12 hpi | Increase 24 hpi | Increase 48–72 hpi | Peptide Family Description |
|---|---|---|---|---|---|
| EPFL | 5/21 | | | ***EPFL1***, ***EPFL14***, *EPFL19*, *EPFL9*, | Epidermal Patterning Factor-Like |
| GASA | 4/28 | *GASA25* | | ***GASA17***, *GASA22*, *GASA29* | Gibberellic Acid Stimulated in Arabidopsis |
| GLV | 3/15 | *GLV9*, ***GLV10*** | | ***GLV8*** | Golven/Root Growth Factor |
| IDA | 1/38 | *IDA15* | | | Inflorescence Deficient in Abscission |
| IMA | 14/14 | | | ***IMA1***, *IMA2*, *IMA3*, ***IMA5***, *IMA6*, *IMA7*, ***IMA8***, *IMA9*, *IMA10*, *IMA11*, *IMA12*, ***IMA13***, ***IMA14***, *IMA15* | Iron Man |
| Kunitz | 2/48 | | | *Kunitz13*, ***Kunitz18*** | Kunitz-P trypsin inhibitor |
| LAT52-POE | 3/40 | | ***LAT52/POE1***, ***LAT52/POE12*** | ***LAT52/POE21*** | LAT52/Pollen Ole e 1 Allergen |
| LCR | 1/89 | | | ***LCR64*** | Low-molecular weight Cys-rich |
| Legin | 8/48 | *Legin20* | | ***Legin32***, *Legin37*, *Legin38*, *Legin42*, *Legin43*, *Legin44*, *Legin47* | Leginsulin |
| LP | 4/20 | | | *LP8*, *LP9*, *LP14*, ***LP15*** | LEED..PEED |
| N26 | 2/4 | | | *N26-3*, *N26-4* | Nodulin26 |
| NCR-A | 13/327 | | | ***NCR025***, ***NCR037***, ***NCR267***, ***NCR279***, ***NCR323***, *NCR376*, ***NCR396***, ***NCR547***, ***NCR639***, *NCR685/NCR686* | Nodule-Specific Cysteine RichGroup A |
| NCR-B | 36/365 | *NCR150* | | ***NCR031***, ***NCR051***, *NCR057*, *NCR117*, ***NCR157***, *NCR158*, *NCR209*, *NCR223*, ***NCR229***, ***NCR235***, *NCR252*, ***NCR308***, ***NCR386***, ***NCR415***, *NCR454*, ***NCR455***, ***NCR465***, ***NCR507***, ***NCR527***, ***NCR529***, ***NCR567***, *NCR568*, *NCR573*, *NCR648*, *NCR657*, *NCR673*, ***NCR678***, ***NCR708***, *NCR713*, ***NCR730***, *NCR736*, ***NCR737***, ***NCR738***, *NCR757*, ***NCR793*** | Nodule-Specific Cysteine RichGroup B |
| NodGRP | 14/54 | *NodGRP15*, *NodGRP45* | | ***NodGRP1B***, *NodGRP3C*, *NodGRP4*, ***NodGRP12***, *NodGRP23*, ***NodGRP30***, *NodGRP32*, ***NodGRP33***, ***NodGRP34***, *NodGRP35*, ***NodGRP36*** | Nodule-Specific Glycine-rich Protein |

**Table 1.** *Cont.*

| Peptide Family | Induced Genes /Total Genes (v4 Genome) | Increase 12 hpi | Increase 24 hpi | Increase 48–72 hpi | Peptide Family Description |
|---|---|---|---|---|---|
| nsLTP | 18/132 | *nsLTP53, nsLTP61, nsLTP62* | *nsLTP100* | *nsLTP25,* ***nsLTP49,*** *nsLTP50, nsLTP51,* ***nsLTP52, nsLTP54,*** *nsLTP72, nsLTP75,* ***nsLTP76, nsLTP81, nsLTP83, nsLTP84,*** *nsLTP102,* ***nsLTP110*** | Non-Specific Lipid Transfer Protein |
| PCY | 11/86 | *PCY16,* ***PCY27,*** *PCY33* | *PCY47,* ***PCY59*** | *PCY19,* ***PCY35,*** *PCY64, PCY68, PCY72,* ***PCY78*** | Plantcyanin/Chemocyanin |
| PDF | 16/16 | *PDF5, PDF6, PDF7, PDF10, PDF13, PDF14, PDF39* | | *PDF2, PDF9, PDF11, PDF27, PDF36,* ***PDF38,*** *PDF44,* ***PDF45,*** *PDF57* | Plant Defensin |
| PSK | 1/10 | | | ***PSK8*** | Phytosulfokine |
| RTFL/DVL | 2/15 | | *RTFL/DVL1* | ***RTFL/DVL13*** | Rotundifolia/Devil |
| STIG-GRI | 1/18 | | | *STIG/GRI4* | Stigma1/GRI |

### 3.3. Rhizobial Response in Shoots

The systemic AON pathway is initiated by the interaction of roots with rhizobia followed by transport of newly synthesized mobile peptide signals to a receptor complex in the shoot. Perception of the signal in the shoots results in a signal sent to roots to inhibit further nodule development. From our DESeq2 pipeline, eleven genes were identified showing a rhizobial response in plant shoots (Figure 7A). Seven genes were induced in both wild-type and AON mutants over the first 72 hpi following rhizobia inoculation, while a single gene was induced in wild-type but not *sunn-4* (Medtr3g111880, a predicted MYB family transcription factor), and three genes were induced in *sunn-4* but not in wild-type (Medtr1g074990, Medtr2g435780, and Medtr4g121913). The seven genes increasing in all three lines include an IRON MAN Peptide gene (IMA11; Medtr4g026440), also found as one of fourteen IRON MAN genes induced by rhizobia in roots, and the gene for carotenoid cleavage dioxygenase 4a-6 (CCD4; Medtr5g025270).

The predicted MYB family transcription factor gene (Medtr3g111880), found to be uninduced in *sunn-4*, increased expression in wild-type by 24 hpi, but by 72 hpi there was still no increase detected in *sunn-4* (Figure 7B). In the *rdn1-2* mutant, expression levels increased but on a slower time frame than in wild-type, with the first increase seen at 48 hpi. This gene was also found to be induced by rhizobia by 48 hpi in roots, where the AON mutants both showed a stronger induction of expression than wild-type.

A predicted sieve element occlusion protein gene (Medtr1g074990) showed an early transient pulse of expression in *sunn-4* that was weaker in *rdn1-2* and absent in wild-type (Figure 7C). Two genes encoding small proteins of unknown function (Medtr2g435780 and Medtr4g121913) showed increasing expression across the time course in *sunn-4* but no apparent increase in *rdn1-2* or wild-type. None of these three genes showed a response in roots, with Medtr4g121913 expression restricted to shoots.

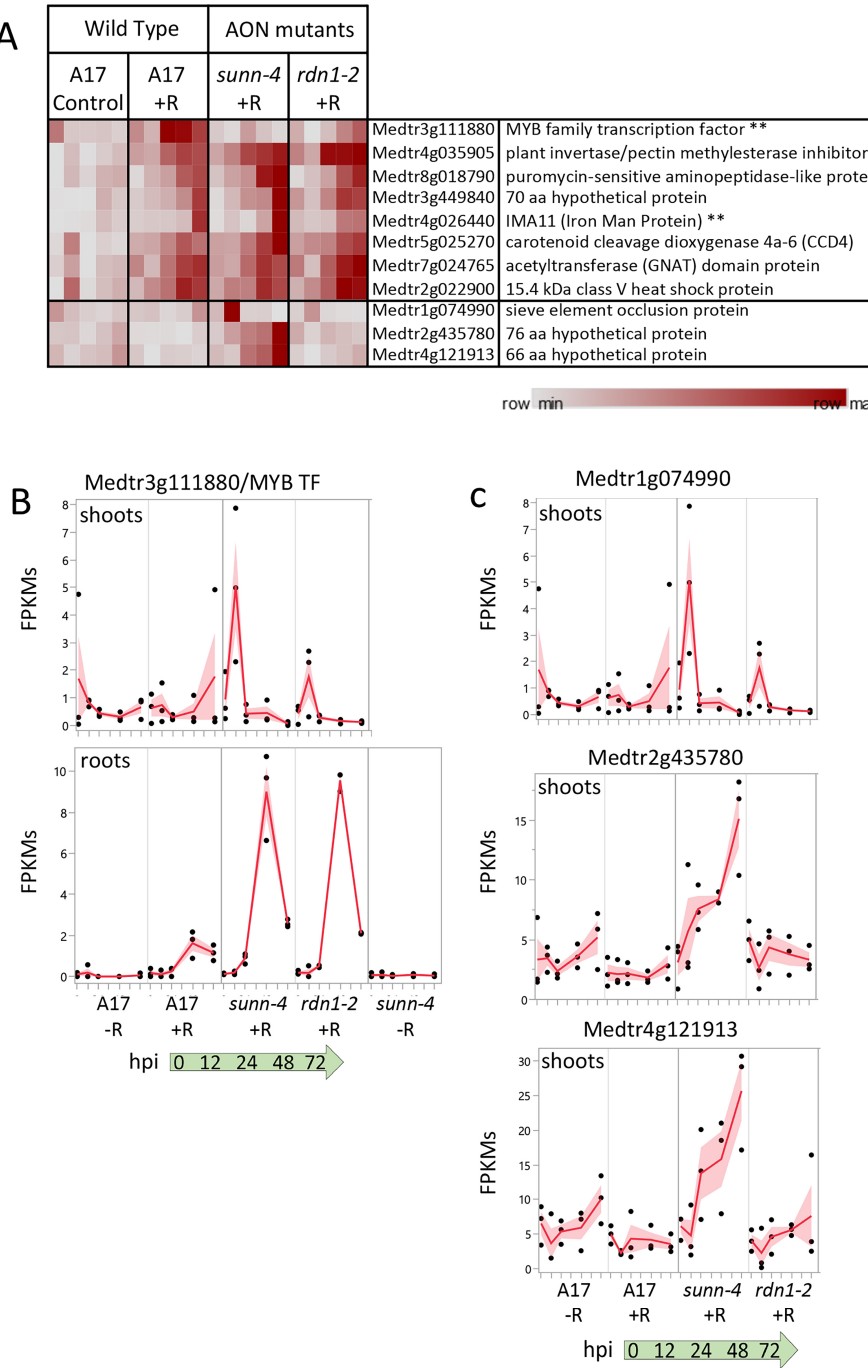

**Figure 7.** Gene expression induced by rhizobia in shoots. (**A**) Heat map of average fragments per kilobase of transcript per million mapped reads (FPKMs) of genes showing increased expression during the first 72 h of nodulation. Each row is independently scaled from minimum to maximum values; underlying data in in Supplemental Dataset 2, Sheet F6. Eight genes were induced in shoots of wild type plants. Seven of these were also induced in shoots of AON mutant plants *sunn-4* and *rdn1-2*. Three additional genes were induced only in *sunn-4*. Two genes that were also induced in nodulating roots are indicated by "**". Graphical representation of selected genes is shown in (**B**,**C**) with FPKMs (black dots) and their means (red lines) from three biological replicates. Shading is the standard error of the mean. (**B**) Transcription factor gene Medtr3g111880 was induced by 24 h in shoots of wild-type but not *sunn-4*, while in roots expression was induced in all three lines. (**C**) Three genes induced in shoots of *sunn-4*.

## 4. Discussion

### 4.1. Expression Differences in AON Mutants

Combining transcriptome data from wild-type and two AON mutants, *sunn-4* and *rdn1-2*, we identified genes with altered expression in roots and shoots of the AON mutants. Using only the part of the root initially responding to rhizobia and following only that part of the root over time in synchronized plants was possible because of the aeroponic system we used to grow the plants. While the simultaneous aeroponic system has been used to generate a transcriptome in a previous study [31] and the use of the area surrounding inoculation has been used on growth plates [32], the two methods have not been combined until this report.

Given the level of molecular communication between the plant and microbe and the critical role of SUNN in AON, we expected that disruption of *SUNN* would significantly impact transcriptomic responses to rhizobia. Intriguingly, most of the differences we found were constitutive and not specific to the rhizobial response, suggesting that the AON pathway is one of multiple signal transduction pathways affected by a mutation in SUNN. A microarray comparison of uninoculated *sunn-1* mutant against the uninoculated *lss* mutants in which no *SUNN* transcript is produced (phenocopy of *sunn-4*) also showed that no *SUNN* expression resulted in less misregulation than mutant *SUNN* expression [33]. While only one nodulation pathway gene from [3], *MtSYT2*, was affected in *rdn-1* mutants and not *sunn-4* mutants (Figure 4), the role of RDN1 in nodulation is predicted to be modification of a ligand of SUNN [12] and the almost complete overlap of mis expression of nodulation genes confirms that.

We identified 54 genes with constitutively altered expression in *sunn-4* plants. Among these are 14 genes with altered expression in both roots and shoots (8 higher, 6 lower), 13 in roots only (7 higher, 6 lower), and 27 in shoots only (10 higher, 17 lower). Expression of 23 of the genes was also altered in *rdn1-2*. An additional seven genes had altered expression in *rdn1-2* only, including one encoding a short hypothetical protein always expressed in the shoots or roots (Figures 2 and 3), and several genes not expressed in the shoots that could be related to disease response (Figure 3). Interestingly, the *NFYC3* transcription factor gene was only expressed in *rdn1-2* roots responding to rhizobia (Figure 2). Since RDN1 is predicted to modify the CLE12 ligand of SUNN [12], these genes that are different in *rdn1-2* mutants and not *sunn-4* mutants could suggest other signaling pathways outside of nodulation that use the modified CLE12 ligand, or other ligands that RDN1 may modify.

Seven of the 54 genes with constitutively altered expression in *sunn-4* plants are predicted to encode proteins of unknown function with lengths ranging from 48 to 236 amino acids. Because they are annotated as hypothetical proteins that could be an artifact of genome annotation, we confirmed that six of the genes have only been identified in *M. truncatula* and the seventh gene has been described in clover. Also among the 54 genes are several members of much larger gene families, including two UDP glycosyltransferases (out of over 250 genes in the family), one legume lectin-binding domain protein (out of 40) and one calmodulin-binding family protein (out of eight). The 4-hydroxy-3 methylbut-2-enyl diphosphate synthase (Medtr2g094160), which is more highly expressed in roots and shoots of both AON mutants, is an enzyme involved in the formation of isoprenoid-derived plant signaling compounds [34]. Other genes misregulated in *sunn-4* included some with suggested functionalities, such as F-box containing genes, transcription factors, and transporters. The collection of genes constitutively altered in *sunn-4* does not point to a known single pathway globally disrupted in these mutants but rather indicates multiple discreet differences for further investigation and suggests possible phenotypes unrelated to nodulation in *sunn-4* plants.

A single gene demonstrated an attenuated rhizobial response in *sunn-4* roots. The isoflavone methyltransferase gene Medtr7g014510, encoding MtIOMT3 [35], was upregulated in wild-type and the *rdn1-2* mutant by 12 h, whereas in *sunn-4* the upregulation was much weaker. This gene is induced in leaves infected with *Phoma medicaginis*, a known inducer of isoflavonoid synthesis [36,37]. MtIOMT3 has been shown to be able to modify a

variety of compounds including 6,7,4′-trihydroxyisoflavone, 7,3′,4′-trihydroxyisoflavone, genistein, glycitein, and dihydrodaidzein [37]. Isoflavones are endogenous regulators of auxin transport in soybean, and genistein production is also part of nodule development in soybean [38]. Since the *sunn-1* mutant has a defect in auxin transport, specifically excess auxin loading in the shoot [39], misregulation of *MtIOMT3* in the roots of *sunn-4* plants could be responsible for some of the auxin transport defect seen in *sunn-1* mutants. However, *sunn-1* mutant roots nodulate normally in composite plants with a wild-type shoot grafted as the shoot [9], making it more probable that *MtIOMT3* is not upregulated because of a *sunn-4* defect in a shoot signal (see Figure 7), suggesting this is a contributor of the phenotype downstream of the shoot signal.

### 4.2. Peptide Responses to Rhizobia

The breadth of the response observed in peptide-encoding genes (Table 1) reflects the ubiquitous nature of peptide function in plant roots [40–42], and members of multiple peptide-encoding gene families were identified in the nodulation response, including CLE peptides with a demonstrated role in nodulation regulation [26,27]. The pattern of peptide-encoding gene response to rhizobia in AON mutants was similar to that in wild-type plants. The main difference was an increased number of peptide-encoding genes induced, mostly at later time points, presumably due to the higher number of nodules present in the analyzed tissue, but there were a few peptide expression patterns of note.

NCR peptides (nodule-specific cysteine-rich peptides), a large group of defensin-like antimicrobial peptides, are produced in nodules of *M. truncatula* and control rhizobial development [43,44]. Interestingly, we found that whereas almost all NCR peptides were nodule-specific, a single NCR peptide gene (*NCR150*) was expressed at our earliest time point, before nodules had developed, and then was turned back off at nodule emergence. NCR150 has previously been shown to be one of the very few NCR peptides known to be expressed outside of nodules, as it was found in epidermal cells following Nod factor treatment [45]. The pattern of expression could be interpreted to imply a role in nodule regulation.

IRON MAN peptides have been shown to be involved in iron transport in arabidopsis [29]. In non-nodulating root samples, *IRON MAN* genes were found to be expressed at nearly undetectable levels, but in nodulating roots all fourteen genes were actively expressed in the AON mutants by 48 to 72 h after inoculation, with nine of them up in wild-type as well. Based on the timing of their induction, it would follow that these genes may be required for signaling root tissue to rapidly synthesize leghemoglobin, which keeps the oxygen tension in the nodule low enough for the bacterial nitrogenase to function. The increased expression levels of IRON MAN peptides in *sunn-4* mutants is likely due to the 10-fold increase in nodules formed, requiring more leghemoglobin.

### 4.3. Rhizobial Response in AON Mutants Includes Induction of TML Genes

Our analysis of expression of functionally validated symbiotic nitrogen fixation genes [3] showed that the majority (over 70%) did not change expression in root tissues during the establishment of nodules. This is not surprising given that many of the genes encode receptors, enzymes, or transporters that may be constitutively expressed or encode proteins that are increased later in nodule development. However, 40 of the 206 genes did have a transcriptional response to rhizobia in wild-type, and all these were also induced in both *sunn-4* and *rdn1-2*. An additional 16 genes had detectable increases following inoculation only in the AON mutants, perhaps because of the higher number of nodules forming on those roots. It is important to note that in our system, nodules are developing and emerging between 48 and 72 hpi. Many of the nodulation pathway genes identified as induced in Figure 4 or Table 1 are associated with the invasion of the nodules by rhizobia (zone II) and are part of the nodule-specific transcriptome [46]. The upregulation detected may be in preparation for or simultaneous with invasion but are downstream in early

signaling. These include SymCRK, NAD1, and DNF2 (Figure 4), and NCRs, LEED..PEEDs, and NodGRPs (Table 1).

Among the genes induced by rhizobia in both wild-type and the AON mutants were *TML2* and *TML1*, encoding two related Kelch-repeat F-box proteins involved in suppressing nodulation [16]. Both genes are targets of miR2111 [47]. It has been proposed that miR2111 transported from shoots to roots maintains nodulation competence by keeping levels of *TML* mRNAs in roots low and that, following nitrate induction of *MtCLE35* [48] or rhizobial induction of *MtCLE13* [45], a SUNN-dependent decrease in miR2111 transport to the roots allows accumulation of *TML* transcripts, proposed to halt further nodule development, leading to the postulate that *sunn* mutants are unable to decrease miR2111 transport, resulting in low TML levels and hypernodulation. Although increased *TML2* and *TML1* gene expression in roots following inoculation with rhizobia is expected in wild-type, the finding that *TML2* was induced in a similar manner up to 24 hpi and *TML1* was induced in the same pattern as wild-type by 72 hpi in *sunn-4* mutants is unexpected. The observed patterns of transcript accumulation for both these genes in the *sunn-4* mutant suggests that early (up to 24 hpi) accumulation is not SUNN-dependent, as no significant difference was seen in this time frame, which may be before the inhibitory signal is processed. Additionally, since *TML1* levels were similar to wild-type up to 72 hpi while *TML2* levels were not, the two *TML*s are likely regulated differently. An analysis of miR2111 accumulation in the roots of wild-type and *sunn-4* mutants up to 48 hpi in the same experimental setup used here was inconclusive (Chapter 4 in [49]), and thus we are unable to correlate miR2111 levels with the *TML* levels observed. The SUNN-dependent increase in *TML2* and *TML1* transcript levels proposed from overexpression experiments with *MtCLE35* in transgenic hairy roots at 14dpi [48] did not measure transcript abundance of the *TML* genes, and measurement of *TML1* and *TML2* abundance in inoculated *sunn* mutants in [46] was performed in one branch of a split root at 5dpi. Our data gathered from a much earlier response to rhizobia in a root section of plants with a single root adds new information to incorporate into the model with further experiments. A recent review noted that other factors in addition to SUNN may contribute to changes in *TML* levels, such as CRA2 signaling [50], and our data support this observation.

## 5. Conclusions

This is the first shoot transcriptome of which we are aware from nodulating *M. truncatula* plants tied to a root transcriptome. The analysis demonstrates that expression of only a small set of genes in both roots and shoots is constitutively altered in the mutant *sunn-4*, despite the dramatic hypernodulation and short root phenotypes, as well as auxin transport differences reported for *sunn* mutants. The remainder of the differences consisted in differences in timing and magnitude of expression. Of genes induced in nodulating roots of wild-type, only a single gene, from a flavonoid synthesis pathway, was found to have a weaker response in *sunn-4*. The early rhizobial response of *sunn-4* included unexpected induction followed by decrease of *TML2* as well as induction of *TML1* similar to wild-type plants, suggesting that early increased expression of these nodulation regulation genes is not sufficient for autoregulation in a *sunn-4* mutant background, a new discovery for AON.

**Supplementary Materials:** The following supporting information can be downloaded at: https://www.mdpi.com/article/10.3390/cimb45060293/s1, Figure S1. Induction of Medtr2g086390 and Medtr1g109600 in nodulating wild-type roots; Figure S2. Transient induction of NCR150; Figure S3. Induction of *MtCLE13* and *MtCLE12* following rhizobial inoculation; Figure S4. Induction of *MtCLE13* and *MtCLE12* following rhizobial inoculation. The graph shows FPKMs (black dots) and their means (red lines) from RNAseq. Shading is the standard error of the mean; Table S1. Primers used in this study; Supplemental Data Set 1. Read mapping and alignment for all libraries in this manuscript; Supplemental Data set 2. Data underlying heat maps, organized in sheets by Figure reference.

**Author Contributions:** Conceptualization, J.A.F., E.L.S. and F.A.F.; methodology, E.L.S., S.A.C. and W.L.P.; software, W.L.P. and Y.G.; validation, E.L.S.; formal analysis, E.L.S. and Y.G.; investigation, E.L.S.

and S.A.C.; resources, J.A.F. and F.A.F.; data curation, E.L.S. and Y.G.; writing—original draft preparation, E.L.S.; writing—review and editing, E.L.S., J.A.F., F.A.F., Y.G., W.L.P. and S.A.C.; visualization, E.L.S. and Y.G.; supervision, E.L.S. and J.A.F.; project administration, J.A.F.; funding acquisition, J.A.F. and F.A.F. All authors have read and agreed to the published version of the manuscript.

**Funding:** This research was funded by the National Science Foundation IOS 1444461 to Frugoli and Feltus and National Science Foundation IOS 2139351 (Frugoli co-PI). The work was made possible in part with support from the Clemson University Genomics and Bioinformatics Facility, which receives support from an Institutional Development Award (IDeA) from the National Institute of General Medical Sciences of the National Institutes of Health under grant number P20GM109094.

**Institutional Review Board Statement:** Not applicable.

**Informed Consent Statement:** Not applicable.

**Data Availability Statement:** The raw data underlying this manuscript is deposited at the National Center for Biotechnology Information under BioProjects PRJNA554677 (inoculated roots; control roots from mutants; inoculated and control shoots) and PRJNA524899 (control roots from wild-type).

**Acknowledgments:** Much of the computation was performed on the Clemson University Palmetto Cluster.

**Conflicts of Interest:** The authors have declared that no competing interests exist for this research work. Leidos and the non-profit Sage Bionetworks have no competing interests for this research work.

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
