# Peer review of "A Medicago truncatula Autoregulation of Nodulation Mutant Transcriptome Analysis Reveals Disruption of the SUNN Pathway Causes Constitutive Expression Changes in Some Genes, but Overall Response to Rhizobia Resembles Wild-Type, Including Induction of TML1 and TML2"

_cimb, doi:10.3390/cimb45060293_

Round 1

Reviewer 1 Report

It is a well-designed and well-performed work on autoregulation of nodulation.

The minor changes I would like to see are related to the discussion of the results:

-Please, indicate in the text that genes (such as SymCRK, NAD1, NCRs...) related to or induced during the invasion of the nodules (zone II) are expressed in this experimental set-up by 72 hpi (Figure 3 and Table 1).

-A bit more discussion on the differences between the rdn and sunn mutants. Maybe more emphasis on the surprising results that very little number of DEGs were observed and the differences might be not related to AON.

Author Response

It is a well-designed and well-performed work on autoregulation of nodulation. 

The minor changes I would like to see are related to the discussion of the results:

-Please, indicate in the text that genes (such as SymCRK, NAD1, NCRs...) related to or induced during the invasion of the nodules (zone II) are expressed in this experimental set-up by 72 hpi (Figure 3 and Table 1).

Thank you for picking up on this.  We have added discussion to lines 400-404 and a reference to the nodule specific transcriptome.

-A bit more discussion on the differences between the rdn and sunn mutants. Maybe more emphasis on the surprising results that very little number of DEGs were observed and the differences might be not related to AON.

This has been added to lines 340-360.

Reviewer 2 Report

Here you can find some of the main mistakes I have detected:

Title:

- Really long, it's necessary to synthesize correctly

Abstract:

- Very cool

Introduction:

- General: Many assessments are vaguely cited

- General: Very short introduction. Some of the topics are badly referenced, very quickly explained or oversee to pass to the next concept

- General: The expression 'see reviews X' or 'reviewed in X' are not correct at all. That's the real meaning to include citations, so it's unnecessary in these kinds of expressions. Moreover, they are causing mistakes in the citation incorporation, like double spaces, wrong orographical marks...

- General: Your contributions to the state-of-the-art by this project cannot be the main part of the introduction. I understand your objectives for the work, but the introductory part is for the people to know the relevant information that drove you to prepare this work. Almost 3/4 are your perspective on the problem, barely some concepts are well introduced here 

- Line 47-50: This process is very important to be defined, but maybe not the best way. Seems very rushy

- Line 53: Be careful with the citations and orthography here.

- Line 55-60: Some of these questions are a little captious or biased. Be careful (quite original way to introduce the topics, tho)

- Line 95: 'sumarized in Materials and Methods'... Yes, of course, I can wait this is happening, why to include this expression here?

- Line 95 and on: Are you describing your methodology approach? And then some of the results? No more revision, rebuild the Introduction section. Some aspects are a fresh way to prepare this section, but in general is far from being correct

Materials and Methods:

- Line 120: Origin of the seeds is necessary

- 'Described in [X]' is not a good way to do citation, and it's not the editorial style of the journal. Need to mention the work (author/s) and then incorporate the citation properly placed. Moreover, this practice is overused, conditioning the understanding of the approaches and the reproducibility. Sometimes with a brief is enough to solve this

- Figure S1A, it's relevant to understand and reproduce the system, use it as supplementary maybe is not the best idea

- Line 130: 150 OD600?!?!?!? How many CFUs is this equivalent to? 10^20? This has no sense for any bacteria culture

- Line 130: If the strains' name has changed, indicate it in brackets, not with a slash. In any case, this process is not very clear...

- Many double spaces, take care

- Line 146: 'prep'... seriously? Not an informal chat here

- Why most of the relevant data was previously published? This is weirdly justified here

- Need to specify the databases where they were deposited

- This whole section requires a rebuilt: little information, lack of sources as the software or technical details required 

Results:

- Line 227-228: Some of they have more expression others have less...this is not relevant at all, is expected, not necessary

- Line 232: This paragraph is discussion, not results

- General: Be careful, many assumptions are not results, but hypothesis and discussion topics

Discussion:

- In general is quite good

Conclusions:

- Consider simplifying, but quite ok

Require a full review, some expressions and way to prepare the text is not correct. Overall is ok

Author Response

Here you can find some of the main mistakes I have detected:

Title:

- Really long, it's necessary to synthesize correctly

 On the reviewer’s advice, we have taken a few words out and rearranged a little but see no further way to synthesize it to shorten. The key points are (1) species transcriptome analysis (2) in the SUNN part of the AON pathway (3) some constitutive changes but mostly normal and (4) one very critical normal expression that was predicted by all models to be disrupted in SUNN. 

Abstract:

- Very cool

 Thank you

Introduction:

- General: Many assessments are vaguely cited

- General: Very short introduction. Some of the topics are badly referenced, very quickly explained or oversee to pass to the next concept.

The template for CIMB emphasizes brevity in each of the requested components of the introduction: purpose of work & significance, state of the research field, aim of the work and highlighting principal conclusions.  We have rearranged the intro to follow this order-the first 3 paragraphs describe the signaling systems we are studying and the state of knowledge. The 4th and 5th paragraph describe the aim of the work and the highlights of the approach and the discoveries. We have moved references so that they do not disrupt the flow of a paragraph which describes knowledge summarized from a review, placing the references at the end of a series of states (we assume this is what “referenced badly” means.) We believe these modifications address the main complaints without ignoring the CIMB request for brevity.

- General: The expression 'see reviews X' or 'reviewed in X' are not correct at all. That's the real meaning to include citations, so it's unnecessary in these kinds of expressions. Moreover, they are causing mistakes in the citation incorporation, like double spaces, wrong orographical marks...

We have removed this statement from the citations,

- General: Your contributions to the state-of-the-art by this project cannot be the main part of the introduction. I understand your objectives for the work, but the introductory part is for the people to know the relevant information that drove you to prepare this work. Almost 3/4 are your perspective on the problem, barely some concepts are well introduced here 

As noted above, we followed the CIMB template instructions but have modified the introduction to  address this as best we can without specific requests to go by.

- Line 47-50: This process is very important to be defined, but maybe not the best way. Seems very rushy

We added specific developmental points at which mutations act on nodulation.

- Line 53: Be careful with the citations and orthography here.

Corrected in rearrangement of sentence.

- Line 55-60: Some of these questions are a little captious or biased. Be careful (quite original way to introduce the topics, tho)

We removed the questions and replaced with specific developmental points at which mutations act on nodulation.

- Line 95: 'sumarized in Materials and Methods'... Yes, of course, I can wait this is happening, why to include this expression here?

Sorry- a previous reviewer asked for this to be added-we are happy to remove it.

- Line 95 and on: Are you describing your methodology approach? And then some of the results? No more revision, rebuild the Introduction section. Some aspects are a fresh way to prepare this section, but in general is far from being correct

 These lines address the “highlighting principal conclusions” requested by the CIMB template-see the note to the general comment above.  

Materials and Methods:

- Line 120: Origin of the seeds is necessary

We grow the seed ourselves and often distribute it to others. We made the mutants long ago from the A17 wildtype seed received from another lab 23 years ago and have propagated in our lab since then (there is no repository for M truncatula seed).  A statement of growth has been added.

- 'Described in [X]' is not a good way to do citation, and it's not the editorial style of the journal. Need to mention the work (author/s) and then incorporate the citation properly placed. Moreover, this practice is overused, conditioning the understanding of the approaches and the reproducibility. Sometimes with a brief is enough to solve this

We removed “described in” and rearranged the sentence but placed the reference at the end; it is a very specific apparatus that affects the results. We do not understand the comment “Sometimes with a brief is enough to solve this”.

- Figure S1A, it's relevant to understand and reproduce the system, use it as supplementary maybe is not the best idea

In a previous version of this manuscript a reviewer suggested adding it as a supplement. We have added it back as Figure 1.

- Line 130: 150 OD600?!?!?!? How many CFUs is this equivalent to? 10^20? This has no sense for any bacteria culture

The statement “150 Optical Density Units” is equivalent to 150 ml of a culture with an OD of 1 (in other words, a large volume culture is spun down and resuspended before being added to the apparatus).  Since an OD600 of 1 = 8 x 108 CFU, we are adding 12 x1010 CFUs.  We have added CFU information for those unfamiliar with this terminology and at the request of another reviewer as well.

- Line 130: If the strains' name has changed, indicate it in brackets, not with a slash. In any case, this process is not very clear...

The strain names have not changed-two genes have been merged into one in the next genome annotation.  We replaced the slash with the word “and” to make this clear.

- Many double spaces, take care

There are no double spaces in the text document we could detect using Microsoft Word Editor.  Perhaps this is from the pdf or this publisher’s or reviewer’s software, as we found the line numbers referred to by this reviewer do not match the document we submitted, suggesting something has occurred between what we submitted and what is being read by the reviewer

- Line 146: 'prep'... seriously? Not an informal chat here

Thank you for noticing-this has been corrected.

- Why most of the relevant data was previously published? This is weirdly justified here.

“Most of the relevant data” is an overstatement-15 of 75 libraries in the root data had been analyzed in a different way in bioinformatics manuscripts on new ways to analyze time course data, but not for the purpose used here. The 60 shoot libraries are first analyzed here. Only 12% of the data has been used in other publications. We thought it important to state that some of it appeared in other places.

- Need to specify the databases where they were deposited.

This is stated in the appropriate section “Data Availability” line 436 in our manuscript.

- This whole section requires a rebuilt: little information, lack of sources as the software or technical details required 

 All software and processes have a citation and we addressed all the technical details specified. Without a specific request we cannot address this further.

Results:

- Line 227-228: Some of they have more expression others have less...this is not relevant at all, is expected, not necessary

Not sure what the reviewer is asking us to do-we said it is not expected that they would all have changes in expression, but some do, and then talked about the changes observed.

- Line 232: This paragraph is discussion, not results

The paragraph in question as far as we can tell is the results of the qRT-PCR of a set of genes, introduced by a description of why we followed up on the transcriptome results for these two genes.  Why is this discussion?

- General: Be careful, many assumptions are not results, but hypothesis and discussion topics

 Without a specific mention of which items the reviewer considers assumptions versus results, we are unable to address this comment.

Discussion:

- In general is quite good

 Thank you.

Conclusions:

- Consider simplifying, but quite ok

We considered, but left it intact due to concerns about being misinterpreted.

Reviewer 3 Report

The manuscript entitled "Transcriptome analysis of Medicago truncatula Autoregulation of Nodulation mutants reveals that disruption of the SUNN pathway causes constitutive expression changes in a small group of genes, but the overall response to rhizobia resembles wild type, including induction of TML1 and TML2." is interesting and can be accepted after some revisions.

Introduction: Line 41; "nitrate poor soil" may be changed to "nitrogen poor soil", because nitrate is not only one form of N in soils.

Line 67; pathway}. pathway).

Line 95; (summarized in Materials and Methods). (details are in Materials and Methods)

Line 115; Please add the abbreviation "hps" after the first appearance of  "hours post inoculation" instead of line 175.

Materials and Methods

Line 119; Plant growth and tissue sampling

Line 130; Please add the microbial density CFU/ml for the inoculum.

Line 146: preparations

Line 198-199; Please show the PCR primers for this experiment.

Results

Please add the full spell of FPKM in the figure legends.

Relatively good, but need to revise the original manuscript.

Author Response

The manuscript entitled "Transcriptome analysis of Medicago truncatula Autoregulation of Nodulation mutants reveals that disruption of the SUNN pathway causes constitutive expression changes in a small group of genes, but the overall response to rhizobia resembles wild type, including induction of TML1 and TML2." is interesting and can be accepted after some revisions.

Introduction: Line 41; "nitrate poor soil" may be changed to "nitrogen poor soil", because nitrate is not only one form of N in soils.

Corrected

Line 67; pathway}. pathway).

As part of rearranging this to remove the phrase “reviewed in” that was objected to by another reviewer, we found this error. Corrected.

Line 95; (summarized in Materials and Methods). (details are in Materials and Methods)

The same other reviewer told us it was not important and incorrect form to refer readers to the materials and methods, so this phrase has been removed.

Line 115; Please add the abbreviation "hps" after the first appearance of  "hours post inoculation" instead of line 175.

Corrected to hpi (assuming the s is a typo)

Materials and Methods

Line 119; Plant growth and tissue sampling

Corrected

Line 130; Please add the microbial density CFU/ml for the inoculum.

Corrected

Line 146: preparations

Corrected

Line 198-199; Please show the PCR primers for this experiment.

Thanks for catching this.  We have added and refer to a supplemental table with all the qPCR primers used in the manuscript.

Results

Please add the full spell of FPKM in the figure legends.

Corrected

Round 2

Reviewer 2 Report

Thanks for the changes, looks better. Still some aspects not well addressed, but I'll leave the editor to decide. Good job

Still think many aspects were not well addressed, but the new version is quite better. On your side then. I feel is ok, but not the best format. The authors claim many journal requirements that could be or not well addressed, but well. Not my story, it's more about editing, I guess